# Structurally Different Exogenic Brassinosteroids Protect Plants under Polymetallic Pollution via Structure-Specific Changes in Metabolism and Balance of Cell-Protective Components

**DOI:** 10.3390/molecules28052077

**Published:** 2023-02-22

**Authors:** Ilya E. Zlobin, Elena D. Danilova, Ol’ga K. Murgan, Liliya V. Kolomeichuk, Raisa P. Litvinovskaya, Alina L. Sauchuk, Vladimir V. Kuznetsov, Marina V. Efimova

**Affiliations:** 1Department of Plant Physiology, Biotechnology and Bioinformatics, Biological Institute, National Research Tomsk State University, Lenin Avenue 36, Tomsk 634050, Russia; 2K.A. Timiryazev Institute of Plant Physiology, Russian Academy of Sciences, Botanicheskaya Street 35, Moscow 127276, Russia; 3Institute of Bioorganic Chemistry, National Academy of Sciences of Belarus, Kuprevich Street 5/2, 220084 Minsk, Belarus

**Keywords:** *Hordeum vulgare* L., polymetallic stress, exogenous and endogenous brassinosteroids, photosystem II, osmotic potential, mineral element contents

## Abstract

Heavy metals and aluminum are among the most significant abiotic factors that reduce the productivity and quality of crops in acidic and contaminated soils. The protective effects of brassinosteroids containing lactone are relatively well-studied under heavy metal stress, but the effects of brassinosteroids containing ketone are almost unstudied. Moreover, there are almost no data in the literature on the protective role of these hormones under polymetallic stress. The aim of our study was to compare the stress-protective effects of lactone-containing (homobrassinolide) and ketone-containing (homocastasterone) brassinosteroids on the barley plant’s resistance to polymetallic stress. Barley plants were grown under hydroponic conditions; brassinosteroids, increased concentrations of heavy metals (Mn, Ni, Cu, Zn, Cd, and Pb), and Al were added to the nutrient medium. It was found that homocastasterone was more effective than homobrassinolide in mitigating the negative effects of stress on plant growth. Both brassinosteroids had no significant effect on the antioxidant system of plants. Both homobrassinolide and homocastron equally reduced the accumulation of toxic metals (except for Cd) in plant biomass. Both hormones improved Mg nutrition of plants treated with metal stress, but the positive effect on the content of photosynthetic pigments was observed only for homocastasterone and not for homobrassinolide. In conclusion, the protective effect of homocastasterone was more prominent compared to homobrassinolide, but the biological mechanisms of this difference remain to be elucidated.

## 1. Introduction

Heavy metals and aluminum are among the most significant abiotic factors that reduce the productivity and quality of crops around the world [1,2,3,4]. The excessive content of mobile forms of aluminum, manganese, and a number of other heavy metals is an inherent property of acidic soils, covering about 30–40% of all arable land, and is one of the most important stress factors for plants on such soils [5,6,7,8,9]. In addition, hundreds of thousands of sites around the world are contaminated with heavy metals as a result of anthropogenic emissions [1]. To date, acid depositions and the use of nitrogen fertilizers result in the expansion of acidic areas [5], while the input of heavy metals into agricultural soils continues as a result of industrial activities and the usage of fertilizers, pesticides, etc. [4]. Thus, the importance of the negative effects of heavy metals and aluminum on plants will likely increase in the future [9].

The polygenic nature of traits linked with plant resistance to heavy metals and with the regulation of their uptake makes difficult the use of breeding or genetic engineering methods to obtain heavy metal-resistant plants, and therefore, the use of growth regulators to control the uptake and resistance of plants to heavy metals is of considerable interest [3]. Brassinosteroids are a group of plant hormones that can increase plant resistance to a wide range of adverse environmental factors. A significant number of studies show that brassinosteroids are able to improve the growth and functional state of plants under conditions of metal-induced stress [10,11]. Moreover, the specific feature of brassinosteroids is their ability to significantly limit the accumulation of heavy metals by plants [3,12]. At the same time, a number of studies have also revealed a negative effect of brassinosteroids on the resistance of plants to heavy metals [2,13]. On the other hand, a number of studies have also found negative effects of brassinosteroids on plant resistance to heavy metals [2,13]. The effects of brassinosteroids on plant resistance to metal-induced stress can strongly depend on the plant developmental stage, experimental conditions, metal species, types, and active concentrations of brassinosteroids used, etc. [2,7,13]. However, the question of whether the chemical structure of brassinosteroids affects the protective responses of plants to heavy metal stress remains open. The most studied brassinosteroids, namely, brassinolide, 24-epibrassinolide and 28-homobrassinolide, possess a lactone structure of the B cycle of the steroid backbone and represent the final link in the biosynthetic chain having high phytohormonal activity. The biological activities of 6-keto brassinosteroids, such as 24-epicastasterone and 28-homocastasterone, which are biogenetic precursors of lactones, are much less studied. Comparative data on the mechanisms of the stress-protective action of ketone- and lactone-containing steroid compounds are virtually absent [14,15].

In this regard, we aimed to compare the effects and stress-protective mechanisms of lactone-containing (homobrassinolide, HBL) and ketone-containing (homocastasterone, HCS) brassinosteroids (Figure 1) on the resistance of barley plants to the polymetallic stress. Barley is highly resistant to various abiotic stressors, but it is sensitive to acidic soils and increases the availability of ions of heavy metals and aluminum in the soil solution [5].

The following tasks were set:-To compare the effects of different methods of HBL and HCS application on the morphological parameters of barley plants under polymetallic stress;-To study the influence of brassinosteroids on the photosynthetic apparatus and the antioxidant system of plants;-To study how hormones influence the accumulation of toxic and essential elements in the roots and shoots of barley plants and the functioning of some systems of detoxification of excessive elements;-To reveal the effect of polymetallic stress on the accumulation of endogenous B-lactones and B-ketones.

## 2. Results

### 2.1. Plant Growth and Morphology

The nutrient medium used to grow plants was characterized by a low pH content, low ionic strength, and a concentration of phosphate ions, thus mimicking the composition of the soil solution (Table 1) [16,17]. As a result, even quite moderate excessive concentrations of heavy metals and aluminum resulted in a substantial reduction in plant growth (Figure 2, Appendix A). The root growth was the most sensitive parameter, decreasing by 38%; the leaf area decreased by 21%; and the stem length decreased by 15%, all of which resulted in a 26% decrease in the fresh weight of plants. HCS had a pronounced protective effect on plant growth (Figure 2, Appendix A) at high HCS concentrations (10 nM) regardless of exposure time. Compared with BS-untreated stress-exposed plants, plants treated with a high concentration (10 nM, regardless of exposure time) of HCS had increased leaf area and total weight, and the length of the stem and root increased under exposure to both low (0.1 nM) and high (10 nM) HCS, regardless of exposure time. The protective effect of HBL was less prominent. Under prolonged (10 days) treatment, both low (0.1 nM) and high (10 nM) HBL concentrations increased total leaf area, whereas stem length was increased only by exposure to high HBL concentrations. The difference in protective effects of HBL and HCS at the same concentration and exposure time could be due to their different bioavailability caused by chemical structure. Therefore, for further analysis, we chose to compare the effects of treatment with HBL and HCS for 10 days at a 10 nM concentration.

### 2.2. Plant Photosynthetic Apparatus

Primary photosynthetic processes were found to be highly resistant to polymetallic stress, with no significant suppression of any parameters studied. The action of brassinosteroids did not influence the photosystem II functioning (Figure 3, Appendix A). However, the polymetallic stress clearly impacted the pigment apparatus of barley plants, decreasing the contents of chl *a* and chl *b,* correspondingly, by 70 and 68% and the carotenoid content by 58% from control (Figure 3, Appendix A). HBL treatment had almost no influence on the pigment contents, whereas HCS treatment increased the contents of chl *a*, chl *b* and carotenoids, correspondingly, to 52%, 46%, and 64% from the control level.

### 2.3. Oxidative Stress and Antioxidant Enzymes

Under the action of polymetallic stress, the content of TBARS increased substantially by 2.37 times in barley roots, whereas no significant effects were observed in the shoots (Figure 4, Appendix A). Brassinosteroids exerted differently directed changes in the TBARS content. HBL decreased the TBARS content in the roots of stressed plants, whereas the shoots’ TBARS content was increased. Treatment with HCS increased the TBARS content in barley roots.

Both polymetallic stress and treatment with brassinosteroids influenced the SOD activity neither in the roots nor in the shoots of barley plants (Figure 4, Appendix A). POD activity increased in the roots of stressed plants by 2.36 times, and were unchanged in the shoots (Figure 4, Appendix A). HCS treatment did not influence the POD activity, whereas HBL induced a more than two-fold increase in the POD activity in the barley roots and shoots compared to the levels in stressed plants.

### 2.4. Accumulation of Mineral Elements

As expected, the conditions of polymetallic stress induced the substantial increase in the contents of all excessive elements in barley roots and shoots. For Ni, Zn, Cu, and Pb, the accumulation was substantially more prominent in the roots than in the shoots (Figure 5 and Figure 6, Appendix A). The extent of metal accumulation was close in the roots and shoots for Mn (by 16.08 and 9.44 times, respectively) and for Cd (by 170 and 256 times, respectively) (Figure 5 and Figure 6, Appendix A). Paradoxically, aluminum was found in substantial concentrations in the organs of control plants; therefore, the measurement of aluminum was likely unreliable. Translocation factors, which are the ratio between shoot and root element contents, decreased in metal-stressed plants for all excessive metals except for Cd, for which the translocation factor increased 1.57 times.

Both brassinosteroids induced the significant decrease in the root contents of all excessive metals except for Cd, for which the effects of hormones were non-significant. The extent of the decrease was similar for both brassinosteroids. In shoots, both HBL and HCS induced the significant decrease in Cd and Zn accumulation, whereas HCS was only effective in decreasing the Mn and Ni accumulation (Figure 5 and Figure 6, Appendix A). For Pb accumulation in the shoot, no effect of HCS was observed, whereas HBL induced the increase in the Pb accumulation in the shoot. Both brassinosteroids did not influence the translocation factors of metals substantially compared to the action of polymetallic stress only.

Compared to the excessive elements, changes in the contents of the remaining elements under polymetallic stress were less prominent. Polymetallic stress induced an increase in the root contents and a decrease in the shoot contents of Na, P, and K, and both brassinosteroids increased Na accumulation in both the roots and shoots. For Mg, polymetallic stress induced the uniform decrease in both the roots and shoots, and brassinosteroids increased the Mg content in both organs.

### 2.5. Expression of Genes of Metal Detoxification

The action of polymetallic stress induced relatively little changes in the expression of all genes studied. The only gene for which more than two-fold changes were observed was *ALMT*, which increased 4.73 times in the roots and decreased by 55% in the shoots of stressed plants (Figure 7A,B, Appendix A). HBL did not exerted significant effects on gene expression in roots, while it increased the expression of all genes in the shoots compared to the levels in stressed plants (Figure 7A,B, Appendix A). HCS decreased the expression of the studied genes in both the roots and shoots.

### 2.6. Accumulation of Endogenous Brassinosteroids

Polymetallic stress did not induce the significant changes in the contents of B-lactones, neither in the roots nor in the shoots of barley plants. The contents of B-ketones remained unchanged in roots, whereas in shoots, a significant 1,9-fold increase was observed compared to control plants (Figure 8, Appendix A).

## 3. Discussion

Brassinosteroids are well-known to exert protective effects on plants threated by heavy metal stress; however, among numerous members of this class, 24-epibrassinolide is by far the most studied protector that forms heavy metal stress, whereas stress-protective effects of other brassinosteroids, including ketone-containing ones, are rarely studied [18]. We observed that HCS exerted a substantial protective effect on the elongation of the roots and shoots of barley plants regardless of the concentrations and exposure time, and at high concentrations (10 nM), it also increased the leaf area and total plant weight. The protective effect of HBL was substantially lower than that of HCS and was generally observed only under a long exposure time (10 days).

The most well-known protective effect of brassinosteroid is their stimulation of a plant’s antioxidative system, and this effect is considered to be the most important component of the protective effects of brassinosteroids under heavy metal stress [3,10,18,19,20]. The stimulating effect of brassinosteroids on the antioxidative system of plants and the resulting decrease in oxidative stress under heavy metal treatment were found in several studies [10,20,21]. However, we did not observe the consistent positive effect of brassinosteroids on the antioxidative system of barley plants (Figure 4, Appendix A). SOD activity was unchanged under polymetallic stress and action of brassinosteroids, whereas POD activity increased under HBL but not under HCS action. The TBARS content was decreased in the roots but increased in the shoots of HBL-treated plants, whereas HCS action increased the root TBARS content. The functioning of the photosystem II, which is well-known for its sensitivity to oxidative damage [22], was not impaired by polymetallic stress and not influenced by hormones (Figure 3, Appendix A). At the same time, we should keep in mind that HCS had a greater protective effect on the carotenoid content under polymetallic stress than HBL. As is known, carotenoids are considered effective quenchers of singlet oxygen (^1^O_2_) and triplet Chls through a physical mechanism that involves the transfer of excitation energy leading to thermal deactivation. Additionally, leaf carotenoids can quench ^1^O_2_ by a chemical mechanism involving their oxidation [23].

To sum up, the positive effects of brassinosteroids were not due to the stimulation of the antioxidative system.

The other possible protective effect of brassinosteroids is their influence on the accumulation of toxic elements (Figure 5 and Figure 6, Appendix A). It was previously shown that the application of 24-epibrassinolide to *B. juncea* L. cv PBR 91 seedlings eliminated the negative effects of heavy metals, such as Zn, Mn, Co, and Ni, improved growth, and reduced the accumulation and absorption of metals [24]. Indeed, brassinosteroids decreased the root accumulation of all excessive metals except for Cd. In shoots, both hormones decreased Cd and Zn accumulation, and HCS additionally decreased Mn and Ni accumulation. In other experiments, the application of HBL regulated Mn uptake and growth of *B. juncea* L. [25] and was involved in GSH metabolism and redox status against Zn toxicity in *R. sativus* seedlings [26]. According to the data obtained, the protective effect of both hormones was not due to the restriction of metal transported to the aboveground part, since the hormones did not substantially influence translocation factors. The effect of both HBL and HCS on the accumulation of excessive metals was quite similar; therefore, it is unlikely that the more pronounced protective effect of HCS on plant growth was due to its better ability to decrease the accumulation of toxic metals compared to HBL. One of the ways that the accumulation of toxic elements can be decreased is via the activation of systems involved in the transport and detoxification of these elements, but we did not observe the prominent stimulating effect of brassinosteroids on the expression of selected genes involved in metal tolerance (Figure 7A,B, Appendix A). On the contrary, HCS decreased the expression of all the genes studied. Therefore, the brassinosteroid-induced decrease in metal accumulation in barley plants was not due to the regulation of the expression of the studied genes.

Considering non-excessive essential elements, the most prominent positive effect exerted by brassinosteroids was the alleviation of the stress-induced decrease in Mg contents, which was complete in the roots, but only partial in the shoots (Figure 5, Appendix A). It is well-known that a Mg deficit leads to a decrease in chlorophyll biosynthesis and therefore to chlorosis [27]. However, it is unlikely that the improved Mg nutrition underlay the HCS-induced increase in the chlorophyll content, since both HCS and HBL induced similar increases in the shoot Mg content, whereas the increase in the chlorophyll content was prominent for HCS but not for HBL (Figure 3, Appendix A).

In conclusion, the protective effect of HCS under the conditions of polymetallic stress was substantially more prominent compared to HBL. The difference between these compounds was neither due to differences in their effects on the antioxidative system nor due to different effects on the accumulation of excessive and essential elements in roots and shoots.

The difference in activity of these two steroid phytohormones could be explained by a specific set of regulating factors for each, which, when acting together, are responsible for the resulting effect. Being dependent on the chemical structure, some molecular characteristics are influenced, and this is expressed in obvious changes in biosynthetic processes, membrane permeability, enzyme activity, and ion transport. At least some of these clearly observed phenomena could be connected with a different affinity of the hormones for the BS-receptor. Another important result pointing to a possible reason for the higher activity of B-ketosteroid hormone compared to B-lactone is our finding a significant activation of BS-ketones biosynthesis under stress, which may be an indication of their protective role under these conditions.

Our results indicate that homocastasterone was more effective than homobrassinolide in mitigating the negative effects of polymetallic stress on plant growth. Both brassinosteroids had no significant effect on the antioxidant system of plants. Both homobrassinolide and homocastron equally reduced the accumulation of toxic metals (except for Cd) in plant biomass. Both hormones improved the Mg nutrition of plants that were treated with metal stress, but the positive effect on the content of photosynthetic pigments, including carotenoids, was observed only for homocastasterone and not for homobrassinolide. However, solving the question of what the molecular mechanisms are the influence of the chemical structure of different groups of brassinosteroids on the plant’s tolerance to polymetallic stress requires further investigation.

## 4. Materials and Methods

The studies were performed on barley plants (*Hordeum vulgare* L.) cv. Biom. Plants were grown in climatic chamber in mixture of soil and perlite for 5 days at a temperature of 19 ± 2 °C, a relative humidity of 50–60%, and a PAR quantum flux density of 200–250 μmol × m^2^ × s^–1^ with a 16 h photoperiod. Then, for 5 days, the plants were adapted to a liquid nutrient medium according to Blamey et al. [16], with pH 4.5. This nutrient medium is characterized by low pH level, low phosphate content, and low ionic strength, and most fully corresponds to the real soil composition of acidified and polluted areas, on which the availability of metals for plants is increased. After the 5 days adaptation period, the experimental period began. Two independent experiments were performed.

In the first experiment, growth parameters of barley plants were analyzed. The experiment lasted for 10 days. The following experimental variants were used:Control variant with the standard nutrient medium.Polymetallic stress by the addition of excessive metal ions to the medium. The effective concentrations of heavy metals (Mn^2+^, Cd^2+^, Cu^2+^, Ni^2+^, Zn^2+^, and Pb^2+^) and aluminum (Al^3+^) were selected on the basis of the typical concentrations of these ions in soil solutions of industrially polluted acidic soils [1,17] and on the basis of the results of our previous experiments [28,29], see Table 1.Pretreatment with 0.1 nM HBL for 1 day by addition to the nutrient medium with the following 10-day polymetallic stress.Pretreatment with 10 nM HBL for 1 day with the following 10-day polymetallic stress.Pretreatment with 0.1 nM HCS for 1 day with the following 10-day polymetallic stress.Pretreatment with 10 nM HCS for 1 day with the following 10-day polymetallic stress.Simultaneous treatment with 0.1 nM HBL added to the nutrient medium and with polymetallic stress.Simultaneous treatment with 10 nM HBL and with polymetallic stress.Simultaneous treatment with 0.1 nM HCS and with polymetallic stress.Simultaneous treatment with 10 nM HCS and with polymetallic stress.

Based on the results of first experiment, the second experiment was performed. Only four experimental variants were used:Control variant with the standard nutrient medium.Polymetallic stress by the addition of excessive metal ions to the medium.Simultaneous treatment with 10 nM HBL was added to the nutrient medium and polymetallic stress.Simultaneous treatment with 10 nM HCS was added to the nutrient medium and polymetallic stress.

The biological material from the second experiment was used for the analysis of plant photosynthetic processes for biochemical and molecular analyses. Photosynthetic activity was determined on intact leaves. For biochemical analyses, root and shoot samples were fixed in liquid nitrogen and freeze-dried under vacuum using a VirTis 6211 sublimation chamber (LabX, Midland, ON, Canada). For RNA isolation, root and shoot samples were fixed in liquid nitrogen at −70 °C.

### 4.1. Physiological and Biochemical Analyses

#### 4.1.1. Determination of Growth Parameters

Root and shoot length, total leaf surface area, and fresh plant biomass were determined. Total leaf surface area was calculated using the Anikeev and Kutuzov formula [30]. Fresh biomass was determined using analytical balance (Sartorius, Goettingen, Germany).

#### 4.1.2. Determination of the Lipid Peroxidation Level

The extent of lipid peroxidation was evaluated by the formation of a colored complex between thiobarbituric acid and thiobarbituric acid-reactive substances (TBARS) upon heating [31]. TBARS content was determined spectrophotometrically using a Genesys 10S UV–Vis Genes spectrophotometer (ThermoScientific, Waltham, MA, USA) at wavelengths of 532 and 600 nm.

#### 4.1.3. Determination of the Photosynthetic Pigments Content

To determine the content of photosynthetic pigments, a sample of leaves (15 mg) was ground in 96% ethanol, and the homogenate was centrifuged for 10 min at 8000 rpm using a MiniSpin centrifuge (Eppendorf, Hamburg, Germany). The optical density of the alcoholic extract (final volume of the solution was 1.5 mL) was measured using a Genesys 10S UV–Vis spectrophotometer (ThermoScientific, Waltham, MA, USA) at 470.0, 648.6, 664.2, and 720.0 nm. The pigment concentration in the alcoholic extract was calculated according to Lichtenthaler [32].

#### 4.1.4. Determination of Photosynthetic Activity

The photochemical activity of plants was measured on a PAM fluorimeter (Mini-PAM II, Heinz-Walz, Effeltrich, Germany). Before measurement, the samples were adapted to darkness for 20 min. Then, the light was switched for 10 min (I = 190 μmol photons m^−2^ s^−1^ PAR). The intensity of the saturating light was 6000 μmol photons m^−2^ s^−1^. Saturating pulses were generated every 30 s. Maximal (F_v_/F_m_) and effective (Y(II)) quantum yields of PSII photochemistry, coefficients of photochemical quenching based on the “lake” model (qL) and the “puddle” model (qP), electron transport rate (ETR), and quantum yields of nonregulated (Y(NO)) and regulated (Y(NPQ)) energy dissipation were calculated using Junior-PAM software WinControl-3.

#### 4.1.5. Determination of the Activity of Antioxidant Enzymes

Total superoxide dismutase (SOD, EC 1.15.1.1) and peroxidase (POD, EC 1.11.1.7) activities were determined in crude extracts of leaf tissues. Leaf samples were ground in liquid nitrogen with insoluble polyvinyl pyrrolidone, extracted in 0.066 M potassium phosphate buffer (pH 7.4) containing 0.5 mM dithiothreitol, and 0.1 mM phenylmethylsulfonyl fluoride in dimethyl sulfoxide, and then centrifuged for 20 min at 8000 rpm and 4 °C using a 5430R centrifuge (Eppendorf, Germany). The1 total SOD activity was determined according to Beauchamp and Fridovich [33]. The reaction medium (2 mL) contained 10 μL of supernatant, 1.75 mL of 50 mM Tris-HCl buffer (pH 7.8), 0.2 mL of 0.1 M DL-methionine, 0.063 mL of 1.7 mM Nitro Blue tetrazolium (Fermentas, Waltham, Massachusetts, USA), 0.047 mL of 1% Triton X-100, and 0.060 mL of 0.004% riboflavin. The reaction proceeded under LED lamps (I = 232 μmol photons/m^−2^ s^−1^) for 30 min. Absorption was measured at 560 nm using a Genesys 10S UV–Vis spectrophotometer (Thermo Scientific, Waltham, MA, USA). Peroxidase activity was determined as previously described [34]. The reaction mixture contained 50 μL of supernatant, 1.95 mL of 0.066 M potassium phosphate buffer (pH 7.4), 200 μL of 7 mM guaiacol, and 250 μL of 0.01 M H_2_O_2_. Absorption was measured at 470 nm using a Genesys 10S UV–Vis spectrophotometer (ThermoScientific, Waltham, MA, USA). The protein content in the samples was determined according to Esen [35].

### 4.2. Analysis of Elemental Composition of Shoots and Roots

The shoot and root samples were digested in glass tubes with solutions of concentrated HNO_3_ and HClO_4_ (2:1 *v*/*v*) for 24 h at room temperature and then incubated in a dry block thermostat at 150 °C for 1.5 h and then at 180 °C for 2 h. To desorb metal ions form the apoplast, roots were rinsed in 2 mM CaCl_2_ before digestion. Analysis of the contents of several elements (Na, Mg, Al, P, K, Ca, Mn, Ni, Cu, Cd, Pb, and Zn) in roots and leaves of barley plants was performed by inductively coupled plasma mass spectrometry (ICP-MS), Agilent 7900, Santa Clara, CA, USA.

#### 4.2.1. Determination of the Endogenous Content of Brassinosteroids

The content of B-lactones (B-lactoneBS) and B-ketones (6-ketoBS) was determined by a two-stage enzyme immunoassay, as described previously [36,37].

#### 4.2.2. RNA Isolation and cDNA Synthesis

Leaf samples were grinded in liquid nitrogen, and RNA was extracted using TRIzol reagent (Invitrogen, Waltham, MA, USA), according to manufacturer’s instructions (http://www.invitrogen.com, accessed on 14 January 2023). Sample contamination with genomic DNA was prevented using DNase I, RNase-free (Thermo Fisher Scientific, USA). Complementary DNA was synthetized with MMLV-RevertAid reverse transcriptase (Thermo Fisher Scientific, USA), using oligo-dT21primer (Evrogen, Moscow, Russia).

#### 4.2.3. Identification of Candidate PCR Reference and Target Genes and Primer Design

Levels of gene transcript content were determined by real-time PCR on LightCycler^®^96 System (Roche, Basel, Switzerland), using SYBR Green I (Eurogen, Moscow, Russia). The reaction mixture contained qPCRmix HS SYBR (Eurogen, Russia), cDNA, PCR primers, and ddH_2_O. Each sample was treated in 3 technical replicates, and ddH_2_O was used as a negative control. Gene-specific primers were selected using Primer-BLAST (https://www.ncbi.nlm.nih.gov/tools/primer-blast/, accessed on 14 January 2023) and Vector NTI 11. The description of genes of interest, primer sequences, amplicone length, and melting temperature are present in Table 2. The following amplification program was used: 95 °C for 5 min, then 40 cycles of 95 °C for 15 s, Tm °C for 30 s, and 72 °C for 15 s. The effectiveness of each primer pair was determined using standard curve, using the serial 10-fold dilution of mixture of cDNA samples. The transcript levels of the genes of interest were normalized to the expression of the *Actin* gene and *HvGAPDH* gene. Transcript amount was calculated according to manufacturer’s instruction using MS Excel.

### 4.3. Statistical Analysis

Each experiment was repeated in at least three biological replicates. The number of plants per biological replicate ranged from 55 to 60. The figures and tables present the mean values and their standard errors (SE). The means were compared with control values at corresponding time points using Student’s *t*-test. Asterisks (*) indicate significant differences from the control variant, and *a* letters indicate significant differences from the polymetallic stress-treated variant (*p* < 0.05).

## Figures and Tables

**Figure 1 molecules-28-02077-f001:**
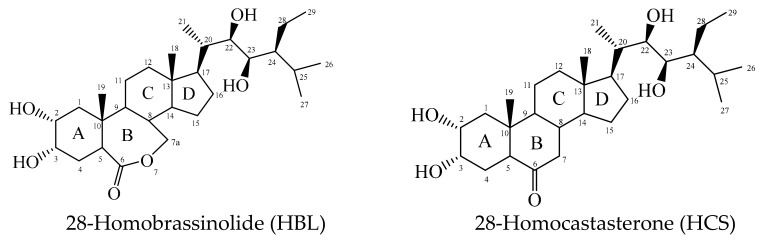
Chemical structure of brassinosteroids.

**Figure 2 molecules-28-02077-f002:**
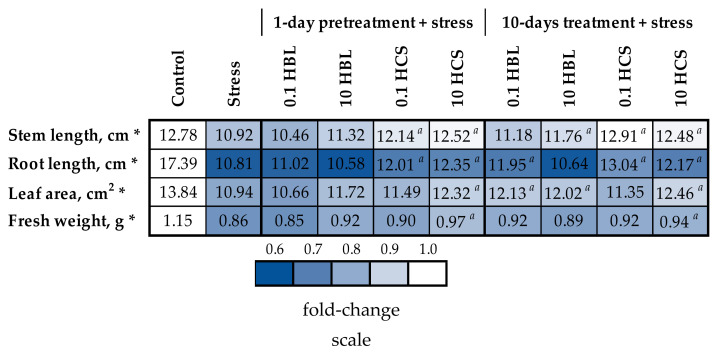
The effects of heavy metal stress and treatment with brassinosteroids on the growth parameters of barley plants. The value in the control plants was taken as 1.0, and the relative decrease is indicated in blue. Here and below: For description of experimental variants, see Materials and methods. Pairwise comparisons of the means were performed using Student’s *t*-test. Asterisks (*) indicate significant differences from the control variant, and the *a* letter indicates significant differences from the polymetallic stress-treated variant (*p* < 0.05).

**Figure 3 molecules-28-02077-f003:**
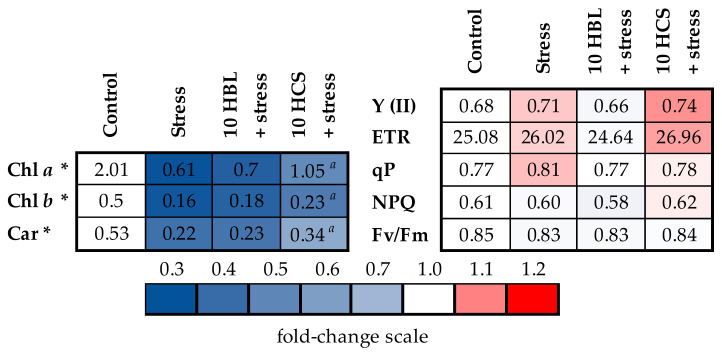
The effects of heavy metal stress and treatment with brassinosteroids on the primary photosynthetic processes and concentrations of photosynthetic pigments (mg/g fresh weight) in barley plants. The value in the control plants was taken as 1.0, the relative decrease is indicated in blue, and the relative increase is indicated in red. Asterisks (*) indicate significant differences from the control variant, and the *a* letter indicates significant differences from the polymetallic stress-treated variant (*p* < 0.05).

**Figure 4 molecules-28-02077-f004:**
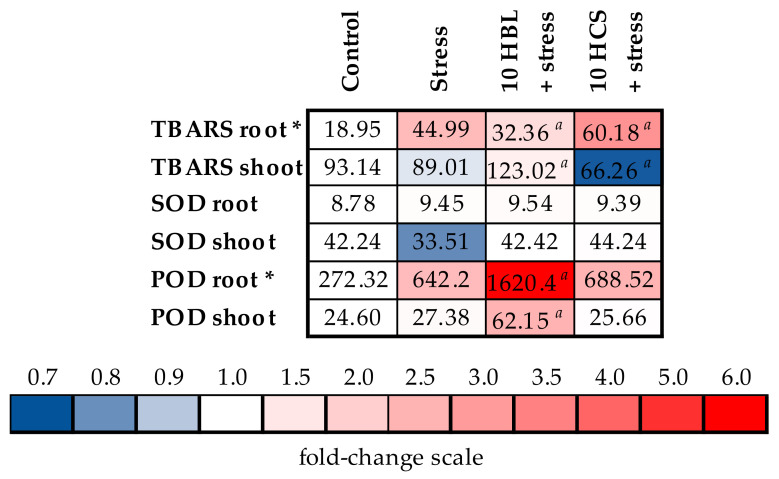
The effects of heavy metal stress and treatment with brassinosteroids on the content of TBARS (ng/g fresh weight) and activity of antioxidant enzymes, SOD (U/g protein) and POD (U/g protein in min), in barley plants. The value in the control plants was taken as 1.0, the relative decrease is indicated in blue, the relative increase is indicated in red. Asterisks (*) indicate significant differences from the control variant, and the *a* letter indicates significant differences from the polymetallic stress-treated variant (*p* < 0.05).

**Figure 5 molecules-28-02077-f005:**
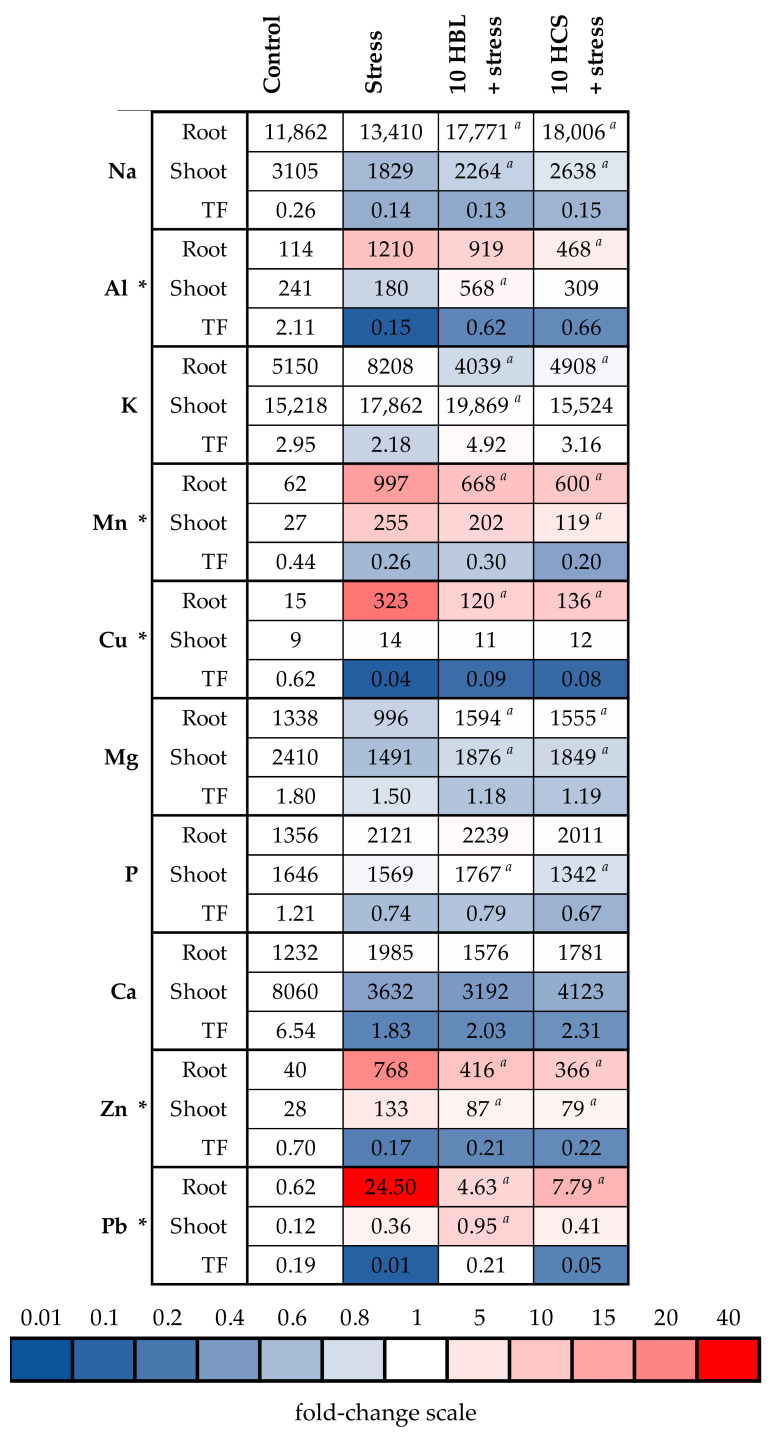
The effects of heavy metal stress and treatment with brassinosteroids on the elemental content (μg/g dry weight) and translocation factors in barley plants. The value in the control plants was taken as 1.0, the relative decrease is indicated in blue, the relative increase is indicated in red. TF—translocation factor. Asterisks (*) indicate significant differences from the control variant, and the *a* letter indicates significant differences from the polymetallic stress-treated variant (*p* < 0.05).

**Figure 6 molecules-28-02077-f006:**
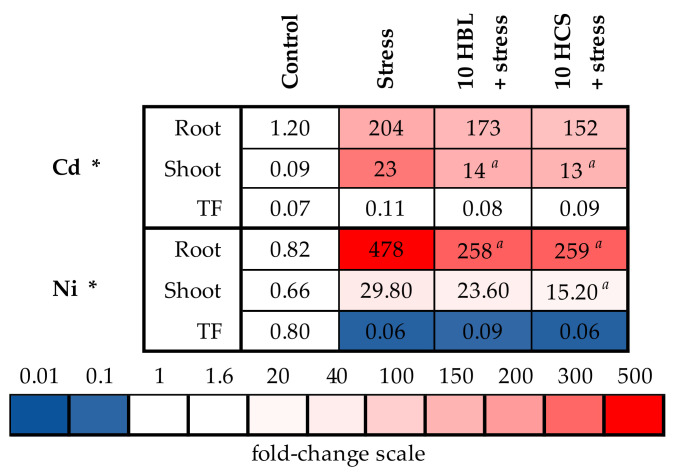
The effects of heavy metal stress and treatment with brassinosteroids on the content of cadmium and nickel ions (μg/g dry weight) and translocation factors in barley plants. The value in the control plants was taken as 1.0, the relative decrease is indicated in blue, the relative increase is indicated in red. TF—translocation factor. Asterisks (*) indicate significant differences from the control variant, and the *a* letter indicates significant differences from the polymetallic stress-treated variant (*p* < 0.05).

**Figure 7 molecules-28-02077-f007:**
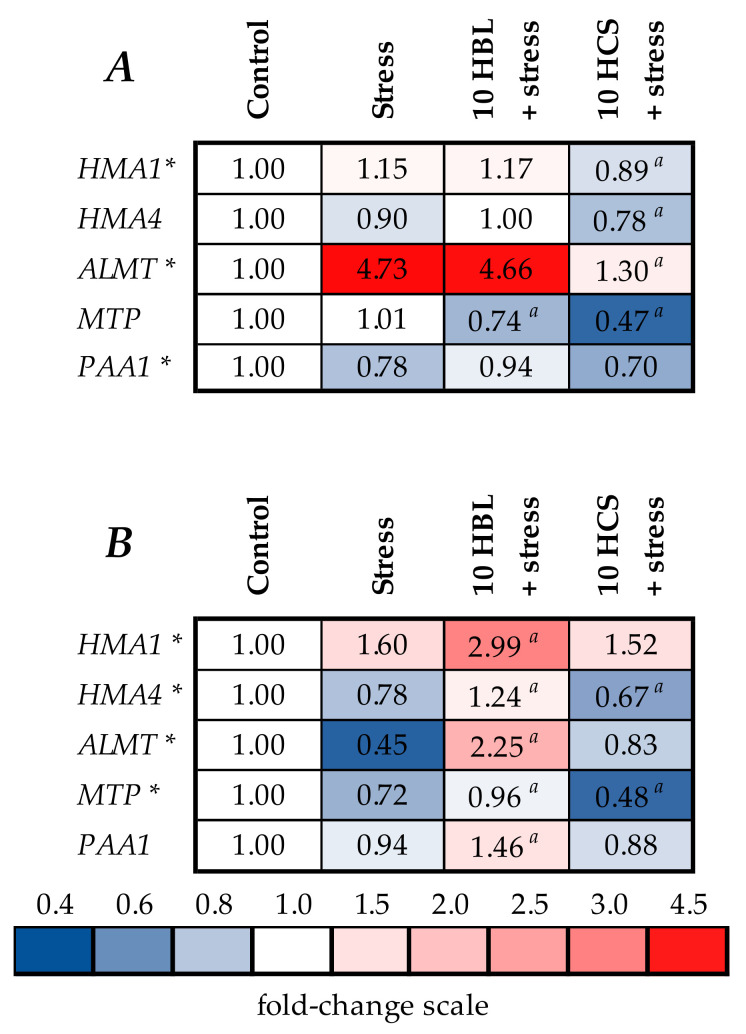
The effects of heavy metal stress and treatment with brassinosteroids on the expression of genes involved in metal detoxification in roots (**A**) and shoots (**B**) of barley plants. The value in the control plants was taken as 1.0, the relative decrease is indicated in blue, the relative increase is indicated in red. Asterisks (*) indicate significant differences from the control variant, and the *a* letter indicates significant differences from the polymetallic stress-treated variant (*p* < 0.05).

**Figure 8 molecules-28-02077-f008:**
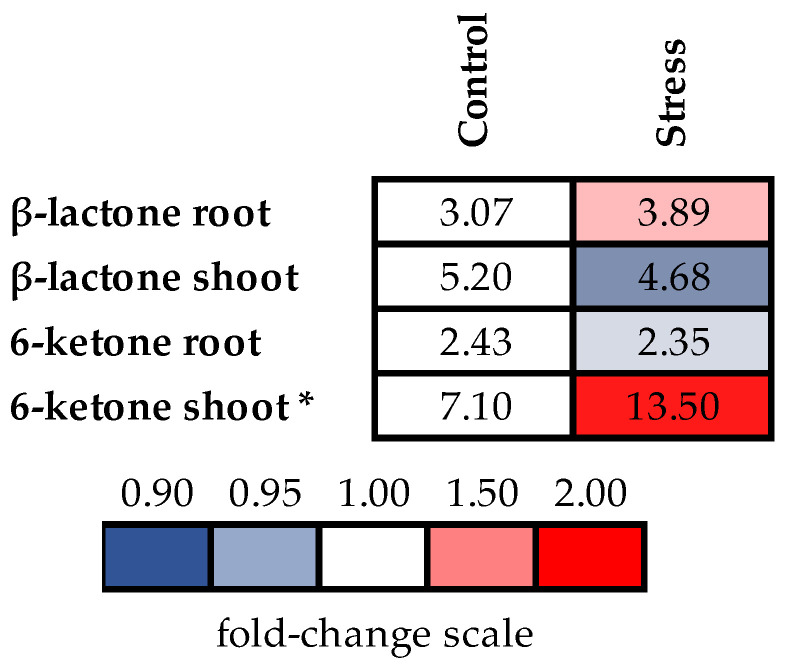
The effects of heavy metal stress on brassinosteroids content (ng/g dry weight) in roots and shoots of barley plants. The value in the control plants was taken as 1.0, the relative decrease is indicated in blue, the relative increase is indicated in red. Asterisks (*) indicate significant differences from the control variant (*p* < 0.05).

**Table 1 molecules-28-02077-t001:** Concentrations of metals.

Metal	Concentration, µM
Al^3+^	20
Mn^2+^	50
Cd^2+^	2.8
Cu^2+^	2
Ni^2+^	16
Zn^2+^	40
Pb^2+^	30

**Table 2 molecules-28-02077-t002:** List of genes, gene-specific primers, and PCR conditions.

Gene ID	Gene	F (5′→3′)	R (5′→3′)	Tm, °C	Amplicon Size, bp	References
LOC123430406	*Actin*, the gene encoding the protein actin	TGGCTGACGGTGAGGACA	CGAGGGCGACCAACTATG	61	121	[38]
LOC123413551	*HvGAPDH*, the gene encoding glyceraldehyde-3-phosphate dehydrogenase 1	GTGAGGCTGGTGCTGATTACG	TGGTGCAGCTAGCATTTGAGAC	61	198	[39]
LOC123406919	*HvPAA1*, encodes a specific ATPase for Cu^2+^/Ag^2+^ transfer	ATGTGCTTGGTCTTGCCA	TCCCTCGCTGTGAGAAGCTA	53	194	[38]
LOC123407761	*HMA1*, encodes a specific ATPase for Zn^2+^/Cu^2+^/Cd^2+^/Pb^2+^ transfer	CCATGTGCATTGGCAGTAGC	AATACATGCCCGCCTTTCAA	59	92 (512)	
LOC123401671	*HMA4*, encodes a specific ATPase forZn^2+^/Cu^2+^/Cd^2+^/Pb^2+^ transfer	GACAGTGGTGGCAGGATTGAAGG	TGGTTCTTGCATCGGTCTCCTCG	64	104	
LOC123414343	*HvMTP1*, encodes a metal resistance protein	CGCAGGATGTGGATGCTGAT	CTCCAGCACCAAAGGCAACA	61	223	[39]
LOC123430267	*ALMT*, encodes the protein aluminum-activated malate transporter	CGGAGCTCTTTGTCGTCAGT	CATTTCCCCACACGCCATTC	60	133	

## Data Availability

The data supporting the findings of this study are available within the article its Appendix A.

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
