# Peer review of "Structurally Different Exogenic Brassinosteroids Protect Plants under Polymetallic Pollution via Structure-Specific Changes in Metabolism and Balance of Cell-Protective Components"

_molecules, 2023, doi:10.3390/molecules28052077_

Round 1
Reviewer 1 Report
The manuscript “Structurally different exogenic brassinosteroids protect plants under polymetallic pollution via structure-specific changes in metabolism and balance of cell-protective components”, compared the performance of two BRs, homocastasterone and homobrassinolide, in barley plants under polymetallic pollution. This paper brings new insight on the biological function of ketone- and lactone-containing steroids. However, I have some questions and suggestions for the authors:
1. For all the multiple data comparison, One-way ANOVA is suggested, and the Standard deviation or standard error should also be provided.
2. No unit of measurement were noted in the Figure1 and Figure 6, please make it clear.
3. A critical problem is the author did not define “translocation factor” at all? What are they? There is no detail methods in the “Materials and Methods” section.
4. How did the authors choose the genes related to metal detoxification without previous reports? please give the detail logic and evidence to choose them.
5. Very poor in the discussion part. The authors need to compare their results to the previous literature about BR derivatives, BR roles in the heavy metal response, and BR’s biological and physical function in barley plants.
6. Use dot i for “0,1 nM” in Line 115, “1,57 times” in Line 175. When you mention a gene of plants, such as “ALMT” make it capital and italic “ALMT”.
Author Response
Responses to Reviewers' Comments
Dear Colleagues,
First of all, the authors sincerely thank both reviewers for their constructive and benevolent critical comments aimed at correcting the deficiencies found in the text. The criticism expressed by the reviewers was extremely helpful, as it significantly improved the manuscript.
Response to Reviewer 1 Comments
Point 1: For all the multiple data comparison, One-way ANOVA is suggested, and the Standard deviation or standard error should also be provided .
Response 1: We agree with the reviewer that a standard error for the mean values should be specified. To this end, six missing tables are included in the Appendix, with the necessary standard errors of the mean values. In the text, references to the tables are highlighted in red.
However, the use of the Student's test in this case seems to us more correct than Anova, because we use pairwise comparison. One-way ANOVA cannot be used, since several factors, namely chemical nature of the hormone, concentration of the hormone and the treatment character (1-day or 10-day) varied.
Point 2: No unit of measurement were noted in the Figure1 and Figure 6, please make it clear.
Response 2: Units have been added to figures 1 and in the description of figure 6.
Point 3: A critical problem is the author did not define “translocation factor” at all? What are they? There is no detail methods in the “Materials and Methods” section.
Response 3: The definition of "translocation factor" has been added to the results section (lines 213-214): «Translocation factors, which is the ratio between shoot and root element concentration…»
Point 4: How did the authors choose the genes related to metal detoxification without previous reports? please give the detail logic and evidence to choose them.
Response 4: The choice of genes of interest is based on the literature data on divalent metal transporters, whose expression increases in response to aluminum and heavy metal stress: Al, ALMT (Delhaize et al., 2012); Cu, Zn, Pb, Cd – HMA1 (Wang et al., 2019), Zn – HMA4 (Zhang et al., 2021), MTP1 (Tiong., 2015), Cu and Cd – PAA1 (Wang et al., 2019 ) and heavy metal resistance proteins – MTP1 (Socha & . Guerinot, 2014).
Delhaize E., Ma J. F., Ryan P. R. (2012). Transcriptional regulation of aluminium tolerance genes. Trends in Plant Science, 17(6), 341–348. doi:10.1016/j.tplants.2012.02.008
Wang X.-K., Gong X., Cao F., Wang Y., Zhang G., Wu, F. (2019). HvPAA1 encodes a P-Type ATPase, a novel gene for cadmium accumulation and tolerance in barley (Hordeum vulgare L.). International Journal of Molecular Sciences, 20(7), 1732. doi:10.3390/ijms20071732
Zhang C., Yang Q., Zhang X., Zhang X., Yu T., Wu Y., Fang Y., Xue D. Genome-wide identification of the HMA gene family and expression analysis under Cd Stress in barley. Plants (Basel). 2021 Sep 6;10(9):1849. doi: 10.3390/plants10091849.
Tiong J., McDonald G., Genc Y., Shirley N., Langridge P., Huang C. Y. (2015). Increased expression of sixZIPfamily genes by zinc (Zn) deficiency is associated with enhanced uptake and root-to-shoot translocation of Zn in barley (Hordeum vulgare). New Phytologist, 207(4), 1097–1109. doi:10.1111/nph.13413
Socha A. L., Guerinot M. L. (2014). Mn-euvering manganese: the role of transporter gene family members in manganese uptake and mobilization in plants. Frontiers in Plant Science. 5. doi:10.3389/fpls.2014.00106
Point 5: Very poor in the discussion part. The authors need to compare their results to the previous literature about BR derivatives, BR roles in the heavy metal response, and BR’s biological and physical function in barley plants .
Response 5: We have tried to enhance the discussion of the results somewhat. It is not easy to do, because there are no data in the literature on the protective effect on plants of different groups of brassinosteroids under polymetallic stress. Changes highlighted in text.
Point 6: Use dot i for “0,1 nM” in Line 115, “1,57 times” in Line 175. When you mention a gene of plants, such as “ALMT” make it capital and italic “ALMT”.
Response 6: The spelling of genes has been changed. Errors in lines 115 and 175 are corrected, highlighted in color. We apologize for the technical errors in the layout of the manuscript.

Reviewer 2 Report
1. What is your aim? add in abstract. Line no. 15 .....grammatical error, ......on acidic and contaminated soils- delete on and add in.
2. In introduction- author hypothesised a question-One unexplored question is how the chemical structure of brassinosteroids influences plant tolerance to heavy metal stress. but conclusion don't have answer of that particular question.
3. Results are clearly presented and have scientific sound. It would be better for public domain if author provide supplementary tables.
4. Discussion is limited. Interpretation is lack. Really results presentation is better but well interpretation is lack.
5. Conclusion-Answer of hypothesised question is missing.
Author Response
Responses to Reviewers' Comments
Dear Colleagues,
First of all, the authors sincerely thank both reviewers for their constructive and benevolent critical comments aimed at correcting the deficiencies found in the text. The criticism expressed by the reviewers was extremely helpful, as it significantly improved the manuscript.
Response to Reviewer 2 Comments
Point 1: What is your aim? add in abstract. Line no. 15 .....grammatical error, ......on acidic and contaminated soils- delete on and add in.
Response 1: The aim has been added to the Abstract and Introduction. (“The aim of our study was to compare the stress-protective effects of lactone-containing (homobrassinolide) and ketone-containing (homocastasterone) brassinosteroids on the barley plants resistance under polymetallic stress”).
The error on line 15 has been corrected. Highlighted in color.
Point 2: In introduction- author hypothesised a question-One unexplored question is how the chemical structure of brassinosteroids influences plant tolerance to heavy metal stress. but conclusion don't have answer of that particular question.
Response 2: Thank you for this comment. In the corrected version of the manuscript this question is posed somewhat differently (“However, the question of whether the chemical structure of brassinosteroids affects the protective responses of plants to heavy metal stress remains open”).
Moreover, in the introduction the tasks we faced are even more specific, including the question posed above.
“The following tasks were set:
- to compare the effects of different methods of HBL and HCS application on the morphological parameters of barley plants under polymetallic stress;
- to study the influence of brassinosteroids on the photosynthetic apparatus and the antioxidant system of plants;
- to study how the hormones influence the accumulation of toxic and essential elements in the roots and shoots of barley plants and the functioning of some systems of detoxification of excessive elements;
- to reveal the effect of polymetallic stress on the accumulation of endogenous B-lactones and B-ketones”.
All of the formulated questions were answered in the course of the study.
Point 3: Results are clearly presented and have scientific sound. It would be better for public domain if author provide supplementary tables.
Response 3: Supplementary tables (S1 – S6) have been added. References to tables in the text of the article are highlighted in color.
Point 4: Discussion is limited. Interpretation is lack. Really results presentation is better but well interpretation is lack .
Response 4: We have tried to enhance the discussion of the results somewhat. It is not easy to do, because there are no data in the literature on the protective effect on plants of different groups of brassinosteroids under polymetallic stress. Changes highlighted in text.
Point 5: Conclusion-Answer of hypothesised question is missing.
Response 5: Thank you. This is indeed an important question. In the course of our work, we tried to compare the stress-protective effects of lactone-containing (homobrassinolide) and ketone-containing (homocastasterone) brassinosteroids on the barley plants resistance under polymetallic stress. In editing the manuscript, we concluded that the chemical structure of brassinosteroids affects the responses of barley plants to polymetallic stress. Specifically, our results indicate that homocastasterone was more effective than homobrassinolide in mitigating the negative effects of polymetallic stress on plant growth. Both brassinosteroids had no significant effect on the antioxidant system of plants. Both homobrassinolide and homocastron reduced equally the accumulation of toxic metals (except for Cd) in plant biomass. Both hormones improved Mg nutrition of plants treated with metal stress, but the positive effect on the content of photosynthetic pigments including carotenoids was observed only for homocastasterone and not for homobrassinolide. However, solving the question of what are the molecular mechanisms of the influence of the chemical structure of different groups of brassinosteroids on the plant tolerance to polymetallic stress requires further serious research.
